# Prompting is not a substitute for probability measurements in large language models

**Jennifer Hu**
Kempner Institute
Harvard University
jenniferhu@fas.harvard.edu

**Roger Levy**
Department of Brain and Cognitive Sciences
Massachusetts Institute of Technology
rplevy@mit.edu

## Abstract

Prompting is now a dominant method for evaluating the linguistic knowledge of large language models (LLMs). While other methods directly read out models' probability distributions over strings, prompting requires models to access this internal information by processing linguistic input, thereby implicitly testing a new type of emergent ability: metalinguistic judgment. In this study, we compare metalinguistic prompting and direct probability measurements as ways of measuring models' linguistic knowledge. Broadly, we find that LLMs' metalinguistic judgments are inferior to quantities directly derived from representations. Furthermore, consistency gets worse as the prompt query diverges from direct measurements of next-word probabilities. Our findings suggest that negative results relying on metalinguistic prompts cannot be taken as conclusive evidence that an LLM lacks a particular linguistic generalization. Our results also highlight the value that is lost with the move to closed APIs where access to probability distributions is limited.

## 1 Introduction

Few technologies have been as exciting — and divisive — for language science as large language models (LLMs). LLMs are capable of incredibly sophisticated linguistic behaviors, which emerge through statistical learning with massive amounts of text data and highly expressive, domain-agnostic learning architectures. On the one hand, the success of these models has sparked a growing movement to treat them as candidate models of human language acquisition and processing (e.g., Baroni, 2022; Warstadt and Bowman, 2022; Wilcox et al., 2022; Contreras Kallens et al., 2023) – indeed, Piantadosi (2023) even claims that they "refute" Chomsky's approach to language. On the other hand, linguists have highlighted shortcomings of

Code and data are available at https://github.com/jennhu/metalinguistic-prompting.

current models that make them unsuitable as cognitive theories (e.g., Dupre, 2021; Lan et al., 2022; Katzir, 2023; Milway, 2023; Murphy, 2023).

No matter their theoretical position, researchers need a way to assess the capabilities of LLMs in order to substantiate such claims. The fundamental unit of LLM computation is $P(token|context)$, which, in principle, can be directly read out from a model by accessing its output layer of vocabulary logits. The distribution that this implies over word strings reflects the model's linguistic generalizations: that is, a generative model of the language seen during training, which can be used to evaluate the likelihood of previously unseen strings. Direct measurements of model-derived string probabilities have revealed capabilities such as syntactic generalizations (e.g., Linzen et al., 2016; Futrell et al., 2019; Hu et al., 2020; Warstadt et al., 2020), semantic plausibility judgments (Kauf et al., 2022), and certain coherence inferences (Beyer et al., 2021).

Recently, there has been a growing trend to use *prompting* to evaluate LLMs' capabilities. Prompting (popularized by Brown et al., 2020) enables end-to-end interaction with models through natural language, and can be done entirely through inference (i.e., without gradient updates). This method has revealed new classes of emergent abilities in LLMs, such as arithmetic, instruction-following, and grounded conceptual mappings (Brown et al., 2020; Wei et al., 2022a; Patel and Pavlick, 2022), as well as the ability to render sentence acceptability judgments (Dentella et al., 2023). From a sociological angle, prompting has also made LLM evaluations more accessible for domain experts in linguistics and cognitive science, sparking new discussion about the limitations of LLMs' abilities (Katzir, 2023; Dentella et al., 2023; Ullman, 2023). For example, Dentella et al. (2023) prompt GPT-3 to produce grammaticality judgments of infrequent linguistic constructions, and conclude that the model "show[s] a critical lack of understanding even of

high-frequency words" (p. 1). Similarly, Katzir (2023) argues that "LLMs are poor theories of human linguistic cognition" (p. 2) based on prompting ChatGPT to compare the well-formedness of two English sentences.

As these examples illustrate, researchers have been making substantial theoretical claims based on prompting. However, there is an important caveat to using prompts to evaluate models' linguistic knowledge. Prompt-based methods test not only whether a model represents the generalization of interest (e.g., a certain ordering of probabilities), but also whether the model can report the outcome of applying the generalization to the sentence in the prompt. In this way, prompting implicitly tests a new type of emergent ability — *metalinguistic judgment* — which has not yet been systematically explored as a way to evaluate model capabilities.

To demonstrate the difference between direct probability measurements and metalinguistic prompting, consider the case of English subject-verb agreement. A direct approach might compare the probability assigned by the model to singular and plural verbs, given a particular subject noun phrase (e.g., Linzen et al., 2016). For example, we might compare $P(\text{"is"})$ and $P(\text{"are"})$, conditioned on the prefix given by (1-a). In contrast, a prompting approach might present a sentence prefix, and pose a question about this linguistic content. For example, we might compare $P(\text{"is"})$ and $P(\text{"are"})$, conditioned on the prompt in (1-b).

(1)     a.     The keys to the cabinet
           b.     Here is a sentence: The keys to the cabinet... What word is most likely to come next?

A model that perfectly performs the metalinguistic task posed in prompt (1-b) should have identical probability distributions over the next word given (1-a) and (1-b). However, there is no guarantee that a model's response to a metalinguistic prompt will match its underlying internal representations. In light of this, how should we interpret models' responses to metalinguistic prompts? How do these responses correspond to models' internal representations? And when should we use metalinguistic prompts as opposed to direct measurements? These questions are becoming increasingly important, as prompting plays a growing role in the debate about LLMs as models of human language processing.

In this study, we do not intend to take a stance on this theoretical debate. Rather, our goal is to evaluate the validity of metalinguistic prompting as a way of measuring LLMs' internal knowledge. We pose two research questions: (1) How well do models perform under direct and metalinguistic evaluation methods? and (2) How consistent are the metalinguistic methods with the direct method? We investigate these questions through four experiments, covering a range of tasks and linguistic domains. Our findings (and their supporting figures) are summarized below:

1. The metalinguistic judgments elicited from LLMs through prompting are *not the same* as quantities directly derived from model representations. *(Figures 2 and 3)*

2. Direct probability measurements generally yield better or similar task performance, compared to metalinguistic prompting. *(Figure 2)*

3. Minimal pairs help reveal models' generalization capacities, compared to isolated judgments. *(Figure 2c vs. Figure 2d)*

4. In general, the less similar the task/prompt is to a direct probability measurement, the worse the alignment between metalinguistic and direct measurements. *(Figure 3)*

Taken together, our findings suggest that negative results relying on metalinguistic prompts cannot be taken as conclusive evidence that an LLM lacks a particular linguistic generalization. These findings suggest a possible basis for a **competence–performance** distinction in LLMs: namely, the distinction between the information encoded in a model's isolated-sentence string probability distribution versus the model's behavioral responses to prompts. We discuss this topic in greater detail in Section 6.1.

Our results also highlight the value that is lost as researchers move toward interacting with LLMs through closed APIs, where access to models' underlying probability distributions is limited. Indeed, only two days before acceptance of this paper to EMNLP, the ability to obtain arbitrary token log-probabilities from gpt-3.5-turbo-instruct was removed from the OpenAI API, reinforcing the timeliness of the issue. We urge future research to clearly motivate the evaluation methods used to assess LLM abilities, and highlight the importance of developing and using open-source models with access to internal probabilities.

| Experiment | Targeted ability | Task | Dataset(s) |
|---|---|---|---|
| 1 (Section 4.1) | Word prediction | Predict final word in a sentence | Pereira et al. (2018); news articles from March 2023 |
| 2 (Section 4.2) | Semantic plausibility | Determine which word (of two options) is most likely, given preceding context | Vassallo et al. (2018) |
| 3a (Section 4.3) | Syntax | Determine which sentence (of two options) is "better", in isolation | SyntaxGym (Hu et al., 2020); BLiMP (Warstadt et al., 2020) |
| 3b (Section 4.4) | Syntax | Determine which sentence (of two options) is "better", given both options | SyntaxGym (Hu et al., 2020); BLiMP (Warstadt et al., 2020) |

Table 1: Overview of experiments in our study.

## 2 Related work

Our study is related to model calibration, or the problem of estimating predictive uncertainty (e.g., Guo et al., 2017; Minderer et al., 2021; Kadavath et al., 2022; Mielke et al., 2022). Most relevant to our work, Kadavath et al. (2022) perform a broad-scale evaluation of "honesty" in LMs. In particular, they analyze models' "truthfulness" and "self-knowledge", which are conceptually similar to our ground-truth and internal consistency evaluations. Similarly, Mielke et al. (2022) find that chatbots' expressions of confidence and doubt are poorly calibrated with the likelihood that the models' responses are actually correct. These approaches differ from ours in that they analyze models' metacognition by annotating verbalized expressions (e.g., "I don't know, but..."), or by using models to evaluate the correctness of their own generated answers.

Other studies have also demonstrated that models can respond to prompts in unexpected ways (Khashabi et al., 2022; Min et al., 2022; Webson and Pavlick, 2022; Webson et al., 2023; Prasad et al., 2023), in some cases achieving successful outcomes even when prompts are misleading or incoherent. Similarly, Turpin et al. (2023) find that explanations elicited through chain-of-thought prompting (Nye et al., 2021; Wei et al., 2022b) can systematically misrepresent the underlying reasons and causes for a model's prediction. McCoy et al. (2023) also demonstrate how LLMs' responses to prompts are highly sensitive to word probabilities, due to their original training objectives.

Beguš et al. (2023) also evaluate LLMs' "metalinguistic" abilities, but in a different sense of the word. They investigate whether LLMs can perform theoretical analyses of the structures and regularities of linguistic expressions — for example, drawing the syntactic tree diagram of a given sentence in valid LATEX code. Beguš et al.'s tested abilities are metalinguistic in the sense that they require judgments about linguistic objects given linguistic input. In contrast, our study investigates whether models can access and report their internal probability distributions through linguistic prompts.

## 3 General methods

### 3.1 Overview of tasks

Our experiments feature four tasks, summarized in Table 1. Together, the tasks cover both word- and sentence-level computations, as well as both isolated judgments and minimal-pair comparisons. Since we aim to analyze the validity of using metalinguistic prompts to reveal linguistic knowledge, the experiments also cover semantic plausibility and syntax as linguistic domains of interest.

As mentioned in Section 1, word prediction (Expt. 1) is a fundamental task for LLMs, and involves the most straightforward operation on models' representations: reading out the logits over the vocabulary in the output layer. Because we know that models represent next-token-probabilities, we can treat the direct probability measurements as providing ground-truth values. We therefore treat word prediction as a baseline task, where we can perform a tightly controlled comparison between probability measurements and metalinguistic prompting. The other three tasks induce an intuitive (partial) ordering in terms of their similarity to the baseline word prediction task (word comparison, sentence judgment, sentence comparison). We return to this ordering in Section 5.2.

### 3.2 Overview of prompts

In each experiment, we evaluate models using four methods: a direct method, and three zero-shot metalinguistic prompting methods. The direct method involves computing probabilities of tokens or full sentences based on the models' internal logits over vocabulary items. In contrast, the metalinguistic

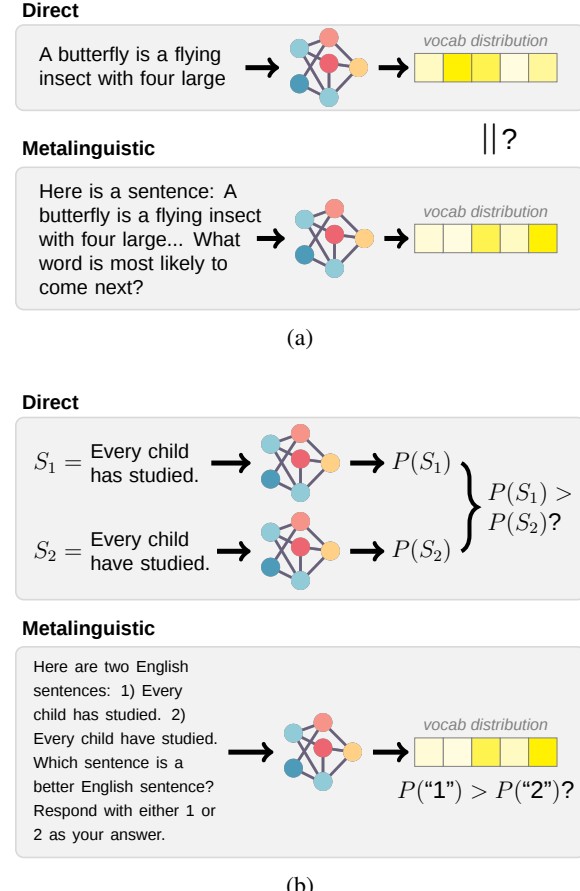

**Direct**

A butterfly is a flying insect with four large → vocab distribution

‖?

**Metalinguistic**

Here is a sentence: A butterfly is a flying insect with four large... What word is most likely to come next? → vocab distribution

(a)

**Direct**

$S_1$ = Every child has studied. → $P(S_1)$

$S_2$ = Every child have studied. → $P(S_2)$

$\left.\begin{array}{c}\\\end{array}\right\}$ $P(S_1) > P(S_2)$?

**Metalinguistic**

Here are two English sentences: 1) Every child has studied. 2) Every child have studied. Which sentence is a better English sentence? Respond with either 1 or 2 as your answer. → vocab distribution

$P(\text{"1"}) > P(\text{"2"})$?

(b)

Figure 1: **Conceptual illustration of direct probability measurements vs. metalinguistic judgments.** (a) Basic word prediction task (Experiment 1, Section 4.1). (b) Sentence comparison task (Experiment 3b, Section 4.4).

prompts ask a question or specify a task requiring a judgment about a linguistic expression.

Taking direct probability measurements as our baseline evaluation method, we also identify an ordering of the three metalinguistic methods in terms of their similarity to baseline. In the MetaQuestionSimple and MetaQuestionComplex prompts, the linguistic object of interest (e.g., a sentence prefix) is linearly closest and farthest from the position where the model is asked to make a prediction, respectively. The MetaInstruct prompts are structured as an imperative instruction, and fall in between the two MetaQuestion* prompts. Tables 2-4b show example prompts for all experiments.[1]

### 3.3 Models

We test six models across all of our experiments: three **Flan-T5** models (small, large, XL; Chung

---

[1]To test the generalizability of our findings beyond English, we also conducted preliminary experiments in Mandarin Chinese. Details and results can be found in Appendix D.

et al., 2022),[2] and three **GPT-3/3.5** models (text-curie-001/GPT-3, text-davinci-002/GPT-3.5, text-davinci-003/GPT-3.5). Flan-T5 models were accessed through Huggingface, and GPT-3/3.5 models were accessed through the OpenAI API.

The Flan-T5 models have an encoder-decoder architecture and were pre-trained on a span corruption task before fine-tuning on a large collection of instruction-based tasks. To measure next-word probabilities under these models, we take the sentence prefix $\langle w_1, w_2, \ldots, w_{n-1} \rangle$ (where $w_n$ is the word to be predicted) and append the sentinel token `<extra_id_0>`. We then create the output sequence `<extra_id_0>` $w_n$, and sum the probabilities corresponding to the tokens of $w_n$. To measure full-sentence probabilities, we compute a pseudo-likelihood inspired by Salazar et al.'s (2020) method for scoring sentences under masked language models. Going left to right, we mask out each word in the sentence using the T5 sentinel tokens, and then sum the log probabilities assigned to each true word.

While the Flan-T5 models differ only in size, the OpenAI models also differ in training regime: text-curie-001 is an autoregressive language model at its core (GPT-3; Brown et al., 2020), whereas text-davinci-002 has additional supervised fine-tuning, and text-davinci-003 has additional reinforcement learning (Ouyang et al., 2022).[3]

## 4 Details of experiments

### 4.1 Experiment 1: Word prediction

**Task and stimuli.** The aim of this experiment is to evaluate models' ability to predict the next word given a preceding context. Instead of a standard language modeling task, where models are evaluated on their ability to predict every word in a text, we use the simplified task of predicting the final word of a sentence. This makes the metalinguistic evaluations more tractable, as we only need to construct a single prompt for each item.

We use two datasets with contrasting style. Our first dataset, taken from Pereira et al. (2018) ("P18"), consists of 384 simple declarative sentences that state a fact about familiar concepts, such

---

[2]Parameter counts: small 80M, large 780M, XL 3B.

[3]Many LLM evaluations use ChatGPT (Piantadosi, 2023; Katzir, 2023) or GPT-4 (Beguš et al., 2023; Webb et al., 2023; Moskvichev et al., 2023). However, we exclude these models from our analyses because the OpenAI API does not provide access to token probabilities for chat-based models.

| Type of prompt | Example |
| --- | --- |
| Direct | A butterfly is a flying insect with four large **wings** |
| MetaQuestionSimple | What word is most likely to come next in the following sentence? A butterfly is a flying insect with four large **wings** |
| MetaInstruct | You are a helpful writing assistant. Tell me what word is most likely to come next in the following sentence: A butterfly is a flying insect with four large **wings** |
| MetaQuestionComplex | Here is the beginning of an English sentence: A butterfly is a flying insect with four large... What is the best next word? Answer: **wings** |

Table 2: Example prompts for Experiment 1. Region where we measure probability is marked in **boldface**. Ground-truth sentence continuations are shown in blue.

| Type of prompt | Example |
| --- | --- |
| Direct | The archer released the {**arrow**, **interview**} |
| MetaQuestionSimple | What word is most likely to come next in the following sentence (arrow, or interview)? The archer released the {**arrow**, **interview**} |
| MetaInstruct | You are a helpful writing assistant. Tell me what word is most likely to come next in the following sentence (arrow, or interview?): The archer released the {**arrow**, **interview**} |
| MetaQuestionComplex | Here is the beginning of an English sentence: The archer released the... What word is more likely to come next: arrow, or interview? Answer: {**arrow**, **interview**} |

Table 3: Example prompts for Experiment 2. Region where we measure probability is marked in **boldface**. Semantically plausible continuations are shown in blue; implausible in red.

as *accordion* or *butterfly*. All pronouns are dereferenced, making the dataset useful for testing prediction in simple contexts with minimal dependencies.

There are at least two concerns with the P18 sentences. First, their simple structure might not be representative of the text that models encounter during training. Second, they have been publicly available since 2018, making it possible that they may be in the models' training data. To address these concerns, we constructed a second dataset ("News") containing sentences that are more naturalistic, but highly unlikely to occur in the training data. We used the NewsData tool[4] to find English news articles published in the United States in the date range of March 20-26, 2023.[5] The articles cover a span of topics, such as business, politics, and food. For each article, we construct a prefix by concatenating the headline with the first sentence (up to, but not including, the last word), separated by the string " – ". There are 222 items in total.

**Prompts.** Example prompts are shown in Table 2 (only examples from the P18 corpus are shown, for simplicity). The Direct prompt feeds the sentence prefix to the model, and we measure the model's probability of the ground-truth next word (indicated

in blue boldface). The other prompts are designed to elicit metalinguistic judgments, through questions (MetaQuestionSimple, MetaQuestionComplex) and instructions (MetaInstruct). As a conceptual illustration, Figure 1a shows a comparison of the Direct and MetaQuestionComplex methods.

**Evaluation.** Our measure of task performance is the log probability assigned by each model to the ground-truth sentence continuation. To measure internal consistency (see Section 5.2), we analyzed the relationship between log probabilities assigned to ground-truth continuations, as measured by the direct method and each metalinguistic method.

### 4.2 Experiment 2: Word comparison

**Task and stimuli.** The aim of this experiment is to evaluate models' ability to judge which of two words is a more likely continuation of a sentence. While Experiment 1 tested word prediction without focus on any particular linguistic phenomenon, here we use the word-comparison task to assess knowledge of *semantic plausibility*.

We use a set of 395 minimal sentence pairs from Vassallo et al. (2018). Each pair consists of two sentences that differ only in the final word, which alters the plausibility of the described event (e.g., "The archer released the arrow/interview"). Each sentence has a simple syntactic structure.

**Prompts.** The prompts are similar to those from Experiment 1, but here we ask models to make a

---

[4]https://newsdata.io/

[5]These are unlikely to be in the Flan-T5 training data, as the models were publicly released in 2022. The OpenAI model documentation also states that text-curie-001 only received training data up to October 2019, and the text-davinci-* models only received training data up to June 2021.

| Type of prompt | Example |
|---|---|
| Direct | {**Every child has studied**, **Every child have studied**} |
| MetaQuestionSimple | Is the following sentence a good sentence of English? Every child has studied. Respond with either Yes or No as your answer. {**Yes**, **No**} |
| MetaInstruct | You are a helpful writing assistant. Tell me if the following sentence is a good sentence of English. Every child has studied. Respond with either Yes or No as your answer. {**Yes**, **No**} |
| MetaQuestionComplex | Here is a sentence: Every child has studied. Is the sentence a good sentence of English? Respond with either Yes or No as your answer. Answer: {**Yes**, **No**} |

(a)

| Type of prompt | Example |
|---|---|
| Direct | {**Every child has studied**, **Every child have studied**} |
| MetaQuestionSimple | Which sentence is a better English sentence? 1) Every child has studied. 2) Every child have studied. Respond with either 1 or 2 as your answer. {**1**, **2**} |
| MetaInstruct | You are a helpful writing assistant. Tell me which sentence is a better English sentence. 1) Every child has studied. 2) Every child have studied. Respond with either 1 or 2 as your answer. {**1**, **2**} |
| MetaQuestionComplex | Here are two English sentences: 1) Every child have studied. 2) Every child has studied. Which sentence is a better English sentence? Respond with either 1 or 2 as your answer. Answer: {**1**, **2**} |

(b)

Table 4: Example prompts for Experiments 3a (a) and 3b (b). Region where we measure probability is marked in **boldface**. Grammatical sentences and correct answer options are shown in blue; ungrammatical/incorrect in red.

comparison between two potential continuations of the sentence prefix (Table 3). For the Direct method, we present the model with the shared sentence prefix, and compare the probability of the plausible (e.g., "arrow"; indicated in blue) and implausible (e.g., "interview"; indicated in red) continuations. For the Meta* prompts, we create two versions of the prompt by shuffling the order in which the answer options are presented.

**Evaluation.** Accuracy is the proportion of items where the model assigns higher probability to the plausible sentence continuation than to the implausible continuation. Random performance is 50%.

To measure internal consistency, for each evaluation method we computed the item-level log probability differentials between the plausible and implausible sentence continuations. We then computed the correlation between the differentials elicited by the direct method and the differentials elicited by each metalinguistic method.

### 4.3 Experiment 3a: Sentence judgment

**Task and stimuli.** The aim of this experiment is to evaluate models' ability to judge whether a sentence is a "good" sentence of English. For any particular sentence, we compare the model's judgment of that sentence to the model's judgment of another sentence that forms a minimal pair, only differing in a critical syntactic feature that manipulates grammaticality or acceptability.

We use minimal pairs from two datasets designed to test knowledge of English syntax: SyntaxGym (Hu et al., 2020; Gauthier et al., 2020), and the Benchmark of Linguistic Minimal Pairs (BLiMP; Warstadt et al., 2020). Since SyntaxGym was not designed for full-sentence probability comparisons, we first converted the SyntaxGym materials into sentence-level minimal pairs.[6] We then took a random sample of 15 items from each of the 23 remaining suites, resulting in 345 items. For BLiMP, we extracted the items that were compatible with the "simple LM method" and then took a random sample of 30 items from each of the 13 categories, resulting in 390 items. See Appendix A for details on the tested phenomena.

**Prompts.** For the Direct method, we measure the probability of each sentence in the minimal pair. For the Meta* prompts, we construct a separate prompt for each sentence in the minimal pair asking whether the sentence is "a good sentence of English" (see Table 4a), and then compare the probability assigned by the model to "Yes" versus "No".

**Evaluation.** To measure accuracy for the direct condition, we compute the proportion of items where the model assigns higher probability to the grammatical sentence in the minimal pair. For

---

[6]We created full sentences by combining content across regions, and then turned each success criterion inequality into a minimal pair (see Hu et al., 2020, for details). For simplicity, we omitted test suites with success criteria involving the conjunction of $\geq 3$ inequalities, or probability differentials.

the metalinguistic prompts, we report balanced accuracy, or the mean of the true positive rate and true negative rate. A true positive occurs when the model assigns higher probability to "Yes" than "No" for a grammatical sentence, and a true negative occurs when the model assigns higher probability to "No" than "Yes" for an ungrammatical sentence.

To measure internal consistency, we compare the log probability differentials for each method. For the direct method, the differential is the difference in log probability of the grammatical and ungrammatical sentences. For each metalinguistic method, the differential is the difference in log probability of the "Yes" token conditioned on the grammatical and ungrammatical sentence prompts.

### 4.4 Experiment 3b: Sentence comparison

**Task and stimuli.** As in Experiment 3a (Section 4.3), the goal is to measure models' syntactic judgments. However, instead of presenting the model with sentences in isolation and asking for judgments, in Experiment 3b we present the model with the minimal pair of sentences, and probe which sentence it takes to be a "better" sentence of English. For our stimuli, we use the same subsets of SyntaxGym and BLiMP as in Experiment 3a.

**Prompts.** The direct evaluation method is the same as in Experiment 3a: we compare probabilities of each sentence in the minimal pair. For the metalinguistic prompts, we have a single prompt for each minimal pair that presents both sentences at once. We assign an identifier (1 or 2) to each sentence in the pair, present a multiple-choice question comparing both sentences, and compare the probabilities assigned by the model to each answer option (i.e., "1" or "2"). As in Experiment 2, we average model results over two versions of the prompt that counterbalance the order of answer options (for metalinguistic prompts). Example prompts are shown in Table 4b. As a conceptual illustration, Figure 1b shows a comparison of the Direct and MetaQuestionComplex methods.

**Evaluation.** Accuracy is measured as the proportion of items where the model assigns higher probability to the grammatical sentence in the minimal pair (direct method), or to the answer option corresponding to the grammatical sentence (metalinguistic prompts). Random performance is 50%.

To measure internal consistency, we compare the log probability differentials between the grammatical and ungrammatical sentences (measured by the direct method) to the log probability differentials between the answer options corresponding to the grammatical and ungrammatical sentences (measured by each metalinguistic prompting method).

## 5 Results

We now return to our main research questions, laid out in the Introduction: (1) How well does each evaluation method perform on each task? (2) How consistent are the metalinguistic evaluation methods with the direct evaluation method? We address these in Sections 5.1 and 5.2, respectively.

### 5.1 Task performance

**Result #1: Metalinguistic judgments are *not the same* as direct measurements.** Figure 2 shows task performance for each experiment. At the coarsest level, the different methods (hues) yield different performance scores, demonstrating that metalinguistic and direct responses are not identical.

**Result #2: Direct measurements generally perform ≥ metalinguistic methods.** Aross all experiments, the direct method nearly always yields best performance of all tested methods. There are a few exceptions: in Experiment 1, Flan-T5-SM performs best under MetaInstruct, and Flan-T5-XL performs relatively well under MetaQuestionComplex, as do the davinci models in Experiment 2.

**Result #3: Minimal pairs help reveal models' generalization capacities.** The difference between Experiments 3a and 3b lies in the presentation of minimal pairs. Comparing Figures 2c and 2d, we first note that the direct results (darkest bars) are identical by definition: they reflect comparisons of full-sentence probabilities. For the metalinguistic prompts, there is an increase in accuracy going from the isolated sentence judgments (Figure 2c) to minimal-pair comparisons (Figure 2d), for all models with above-chance performance.

### 5.2 Internal consistency

**Result #4: Consistency gets worse as we get further from direct measurement of next-word probabilities.** Figure 3 illustrates alignment between direct and metalinguistic measurements (see Appendix Figure 6 for by-model internal consistency results). Each cell shows the average correlation coefficient (Pearson's $r$) between the item-level differentials measured by the direct method

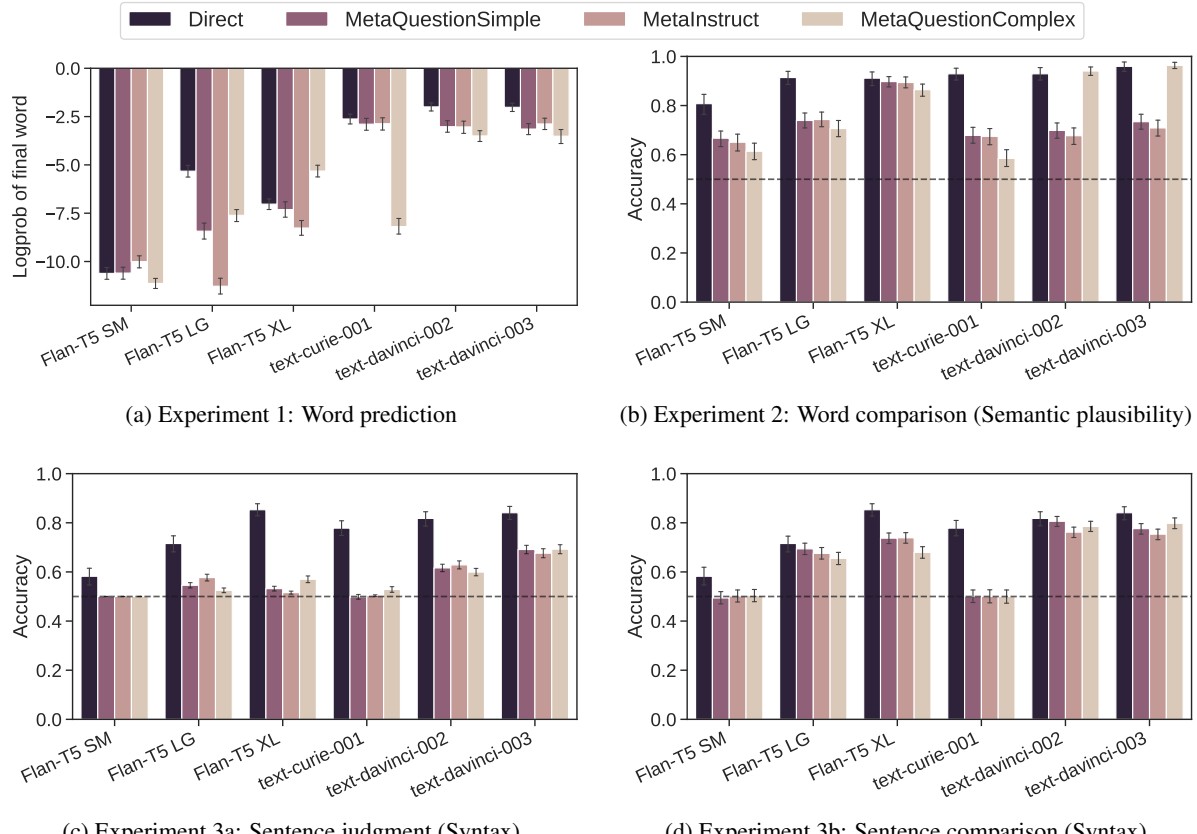

(a) Experiment 1: Word prediction

(b) Experiment 2: Word comparison (Semantic plausibility)

(c) Experiment 3a: Sentence judgment (Syntax)

(d) Experiment 3b: Sentence comparison (Syntax)

Figure 2: **Task performance: Direct probability measurements generally outperform metalinguistic prompts.** (a) Log probability assigned to ground-truth sentence continuation, averaged over items and datasets. (b) Proportion of items where model prefers semantically plausible continuation over implausible continuation. (c)-(d) Proportion of items where model prefers grammatical sentence over ungrammatical sentence in minimal pair, averaged over datasets. Error bars denote bootstrapped 95% CIs. Dashed lines indicate random baseline.

and a particular metalinguistic prompting method.[7]

Recall from Section 3 that the tasks and prompts in our study induce intuitive orderings in terms of how similar they are to word prediction (at the task-level) and direct probability measurements (at the prompt-level). The columns of Figure 3 (prompt types) are loosely ordered by similarity to direct probability measurements, and the rows (tasks) are loosely ordered by similarity to word prediction. Broadly speaking, we find that as distance in either dimension increases, the correlations get weaker. These results further support Result #1: while direct and metalinguistic responses are highly correlated for some combinations of tasks and prompts, the relationship is far from perfect.

## 6 Discussion

In this study, we compared metalinguistic prompting and direct measurements as ways of evaluating LLMs' linguistic knowledge. Broadly, we find that metalinguistic judgments are inferior to direct measurements of token- or sentence-level probabilities. We also find evidence that minimal-pair comparisons help reveal models' generalization capacities.

We do not intend to claim that prompting should be categorically dispreferred in favor of other evaluation methods. Prompting is useful for eliciting open-ended responses, such as chain-of-thought reasoning. Metalinguistic prompting could also be used to ask questions that would be challenging to translate into direct probability measurements (e.g., "Which of the following two sentences is more semantically plausible, but less syntactically well-formed?"). In addition, prompting lowers the technical barrier for domain experts to conduct LLM evaluations, which could contribute to the development of more robust behavioral benchmarks.

What do our findings mean for researchers interested in the linguistic abilities of LLMs? While there are valid reasons to prefer both prompting

---

[7]For Experiment 1, the correlation is computed between the item-level probabilities of the ground-truth final word.

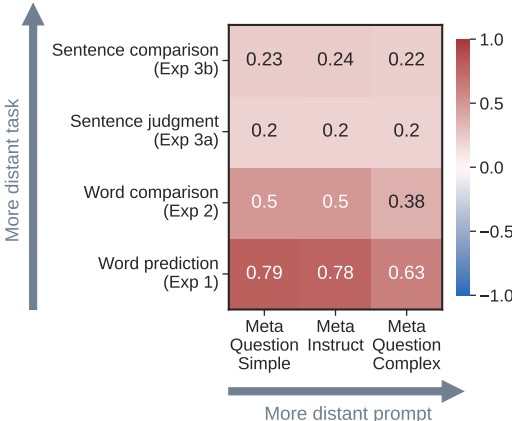

Figure 3: **Internal consistency: Correlation between metalinguistic and direct responses gets weaker as prompts become less direct.** Pearson $r$ correlation between response magnitudes (averaged over models and datasets) measured by direct prompts versus each metalinguistic prompt. See Appendix C for more details.

and direct probability measurements, these methods will not necessarily generate consistent results. More specifically, our findings suggest that negative results relying on metalinguistic prompts cannot be taken as conclusive.

While our paper focuses on the value of direct probability measurements for evaluating linguistic generalizations, other endeavors in cognitive science and machine learning also rely on access to LLM probabilities. For example, token probabilities enable multiple-choice evaluation and beam search, as well as investigating mode collapse (Janus, 2022) and obtaining quantities for Bayesian inference (e.g., Choi et al., 2022; Li et al., 2023; Lipkin et al., 2023).[8] Thus, the value that is lost as we move toward closed APIs extends beyond linguistic analysis, further underscoring the importance of using and developing LLMs with open access to internal probabilities.

### 6.1 Competence vs. performance in LLMs

Humans use language in diverse contexts with varying task demands and constraints, but the underlying linguistic knowledge is relatively stable. This is the **competence–performance** distinction (e.g., Yngve, 1960; Chomsky, 1965): an individual's performance in a particular context may not reflect an individual's full underlying competence. Whether it is productive to distinguish between

"competence" and "performance" in an AI model has been a topic of debate. Katzir (2023) argues that "[LLMs'] behavior directly reflects their competence, and when they fail it is their competence that is at fault" (pp. 4-5). We take Katzir to mean that an LLM's next-word probability distribution is deterministic given its architecture, weights, and the preceding context, and this probability distribution is always computed when the LLM is used. Therefore, as Katzir says, while humans may recover from linguistic errors (e.g., in agreement or parsing) given additional time or resources, "further time and resources are of no use" to LLMs. Firestone (2020) and Lampinen (2023), in contrast, argue that performance conditions need to be carefully controlled in both humans and machines in order to make fair comparisons between the two.[9]

Our work suggests that if a competence–performance distinction is to be made for LLMs, a natural locus is the contrast between the information contained in an LLM's string probability distribution (corresponding to its competence) versus the behavior the LLM exhibits when prompted (corresponding to its performance). A model's failure to exhibit a linguistic generalization when prompted might not reflect a lack of the relevant information in its underlying conditional probability distributions, but instead an inability to access and behave in accordance with that information in response to a prompt that poses a metalinguistic query. This view remains consistent with the fact that LLM behavioral errors may be corrected when illustrative examples are included in the prompt — whether or not such prompts pose metalinguistic queries, they offer opportunity for in-context learning (Brown et al., 2020) — or by allowing the model to produce a reasoning trace before outputting an answer (Nye et al., 2021; Wei et al., 2022b; Kojima et al., 2022).

To conclude, prompting is not a substitute for direct probability measurements in LLMs. We underscore the importance of specifying the assumptions underlying methodological choices in LLM evaluation, and using open models with direct access to probabilities for scientific research. If our interactions with LLMs are limited to high-level prompting, we lose the opportunity to measure capabilities that could advance our understanding of these models and their relation to human language.

---

[8]These examples were inspired by discussion by Tan Zhi Xuan on Twitter (https://twitter.com/xuanalogue/status/1637302507389984769).

[9]Hahn et al. (2022), for example, show that imposing performance constraints on LLMs (by degrading context representations) derives human language processing behavior in complex nested dependency constructions.

## Limitations

Our experiments only test three types of metalinguistic prompts, and only perform zero-shot evaluations. In practice, the small number of metalinguistic prompt types was sufficient to illustrate the difference between direct and metalinguistic responses. However, it would be beneficial to consider more types of prompts to determine how well the phenomenon generalizes. We also note that models might achieve better task performance under the metalinguistic prompts with few-shot prompting or in-context learning. We did not include few-shot analyses due to space limitations, and because many recent LLM evaluations rely on zero-shot metalinguistic prompts. It remains to be seen whether metalinguistic and direct responses are better aligned when models have access to examples in the prompt.

Another limitation of our study is that we only tested a small class of models. An important direction for future work would be to replicate our experiments on models of different sizes and training objectives (e.g., chat-based models). We also note that the results from the OpenAI models are not necessarily reproducible due to the models being behind a closed API. Timestamps of our calls to the OpenAI API are available in our data files (https://github.com/jennhu/metalinguistic-prompting).

## Ethics Statement

This work does not release any new models or datasets. Instead, the goal is to provide insights into methodology for evaluating the internal knowledge of modern LLMs, and in turn contribute to the interpretability of these models. We hope that our results illustrate the importance of open access to model representations and the risks of relying on high-level API interactions for scientific research.

With that said, the broader ethical concerns about LLMs are still relevant to our work. LLMs have been shown to produce output that is factually incorrect, offensive, or discriminatory, and should therefore be used with extreme caution, especially in commercial applications or user-facing settings. Any demonstrations of LLMs' linguistic generalizations should not imply that they are safe to use, or can be expected to behave in a way that is aligned with human preferences and values.

## Acknowledgements

We thank Jon Gauthier and the anonymous reviewers for feedback on earlier versions of this work, as well as Peng Qian for translating the prompts into Chinese. J.H. gratefully acknowledges support from an NSF Graduate Research Fellowship (#1745302), an NSF Doctoral Dissertation Research Improvement Grant (BCS-2116918), and the Simons Center for the Social Brain at MIT. R.P.L. gratefully acknowledges support from NIH grant U01-NS121471, a Newton Brain Science award, and the Simons Center for the Social Brain at MIT.

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

## A Details of syntactic phenomena

Table 5 summarizes the syntactic phenomena covered in the datasets used in Experiments 3a and 3b, which were taken from SyntaxGym (Hu et al., 2020; Gauthier et al., 2020) and BLiMP (Warstadt et al., 2020).

## B Dataset-level task performance

Figure 4 shows task performance for each of the tested datasets in Experiment 1 (left: P18; right: News). Figure 5 shows task performance for each of the tested datasets in Experiments 3a and 3b (left: SyntaxGym; right: BLiMP).

## C Relationship between metalinguistic and direct predictions

Figure 6 shows the average Pearson $r$ correlations between metalinguistic and direct responses, for each of the tested models.

Figures 7 to 10 show the relationship between responses measured by the direct and metalinguistic prompting methods, for Experiments 1-3b, respectively.

## D Experiments in Mandarin Chinese

To explore the robustness of our results beyond English, we performed a preliminary investigation of GPT-3.5 (text-davinci-003) on materials in Mandarin Chinese. We first consulted a native speaker to translate the metalinguistic prompts into Chinese. We then tested GPT-3.5 on two datasets: (1) a set of recent news articles for word prediction, mirroring Experiment 1; and (2) a set of controlled minimal pairs that cover semantic and syntactic phenomena (Wang et al., 2021), mirroring a combination of the phenomena tested in Experiments 2 and 3. We tested the tasks of Experiments 3a and 3b on this set of minimal pairs.

Figure 11 shows the results (task performance) for Experiments 1, 3a, and 3b in Chinese. Like our English experiments, in all our Chinese experiments we find that metalinguistic prompting and direct probability measurements do not yield the same results. Like our Experiment 1 in English, we also find that in the Chinese word prediction task, GPT-3.5 assigns highest probability to the ground-truth continuation under the "Direct" method. We also find that model performance improves when using minimal pairs, mirroring our original findings comparing Experiments 3a and 3b. These findings demonstrate how our English results may generalize to languages with different syntactic structures and grammatical properties.

However, we also found differences between the Chinese and English results: in the Chinese version of Experiment 3b, the "Direct" method underperformed the metalinguistic methods (accuracy scores: Direct 0.6; others 0.8). One potential explanation for this is that according to the OpenAI documentation, the models are "optimized for use in English," although in practice they may work well for other languages.[10] Therefore, the models may not be well-suited to scoring probabilities of Chinese sentences with no context (as in the Direct condition); instead, they may benefit from seeing additional Chinese text in the prompt before the sentence (as in the metalinguistic conditions).

---

[10]https://help.openai.com/en/articles/6742369-how-do-i-use-the-openai-api-in-different-languages

| Dataset | Phenomenon | # items |
|---|---|---|
| SyntaxGym | Center embedding | 15 |
| SyntaxGym | Center embedding (modifier) | 15 |
| SyntaxGym | Cleft | 15 |
| SyntaxGym | Cleft (modifier) | 15 |
| SyntaxGym | Filler-gap dependencies (3 sentential embeddings) | 15 |
| SyntaxGym | Filler-gap dependencies (4 sentential embeddings) | 15 |
| SyntaxGym | Filler-gap dependencies (hierarchy) | 15 |
| SyntaxGym | Filler-gap dependencies (object extraction) | 15 |
| SyntaxGym | Filler-gap dependencies (prepositional phrase extraction) | 15 |
| SyntaxGym | Filler-gap dependencies (subject extraction) | 15 |
| SyntaxGym | Subject-verb number agreement (object relative clause) | 15 |
| SyntaxGym | Subject-verb number agreement (prepositional phrase) | 15 |
| SyntaxGym | Subject-verb number agreement (subject relative clause) | 15 |
| SyntaxGym | Reflexive number agreement (object relative clause, feminine) | 15 |
| SyntaxGym | Reflexive number agreement (object relative clause, masculine) | 15 |
| SyntaxGym | Reflexive number agreement (prepositional phrase, feminine) | 15 |
| SyntaxGym | Reflexive number agreement (prepositional phrase, masculine) | 15 |
| SyntaxGym | Reflexive number agreement (subject relative clause, feminine) | 15 |
| SyntaxGym | Reflexive number agreement (subject relative clause, masculine) | 15 |
| SyntaxGym | Subordination | 15 |
| SyntaxGym | Subordination (object relative clause) | 15 |
| SyntaxGym | Subordination (prepositional phrase) | 15 |
| SyntaxGym | Subordination (subject relative clause) | 15 |
| BLiMP | Anaphor agreement | 30 |
| BLiMP | Argument structure | 30 |
| BLiMP | Binding | 30 |
| BLiMP | Control raising | 30 |
| BLiMP | Determiner-noun agreement | 30 |
| BLiMP | Ellipsis | 30 |
| BLiMP | Filler-gap dependency | 30 |
| BLiMP | Irregular forms | 30 |
| BLiMP | Island effects | 30 |
| BLiMP | NPI licensing | 30 |
| BLiMP | Quantifiers | 30 |
| BLiMP | S-selection | 30 |
| BLiMP | Subject-verb agreement | 30 |

Table 5: Coverage of syntactic phenomena in stimuli used in Experiments 3a and 3b.

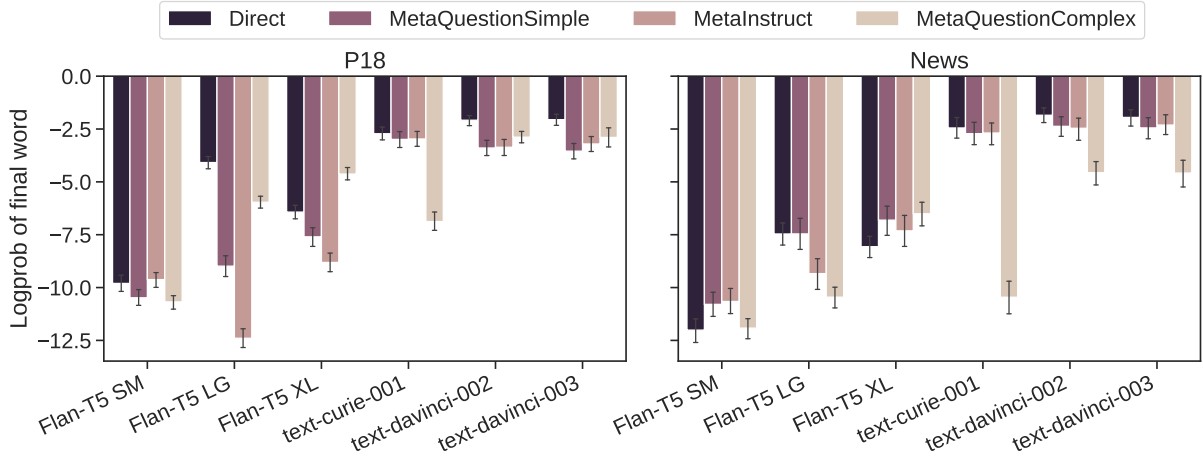

Figure 4: Task performance for each tested dataset in Experiment 1 (word prediction).

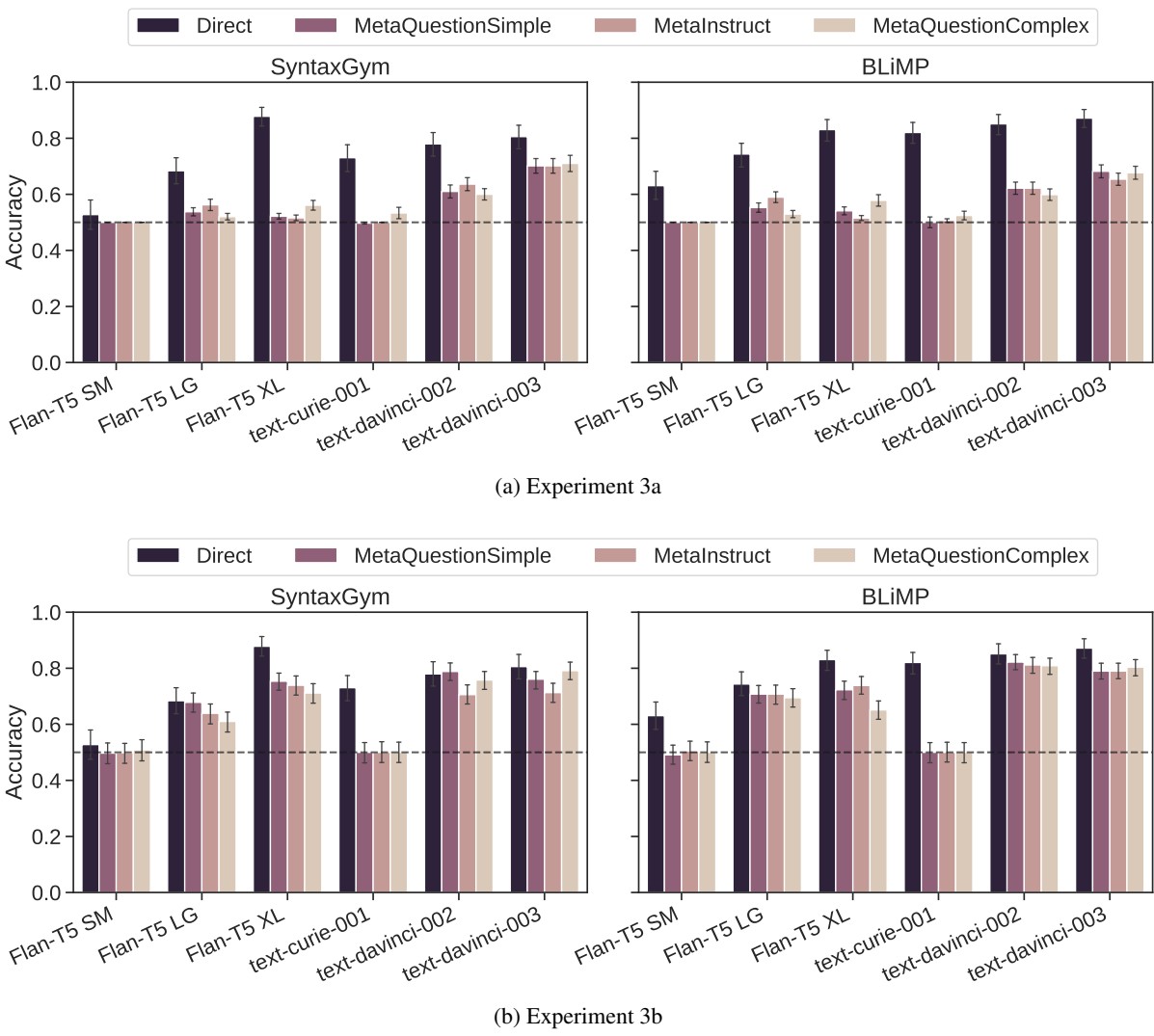

(a) Experiment 3a

(b) Experiment 3b

Figure 5: Task performance for each tested dataset in Experiments 3a (top) and 3b (bottom).

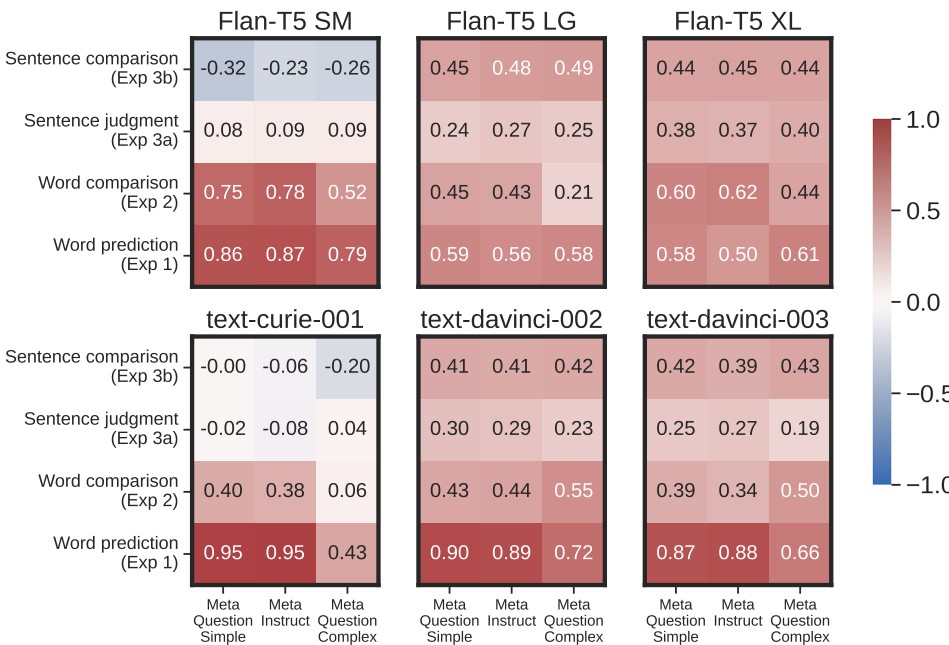

Figure 6: **Internal consistency: Correlation between metalinguistic and direct responses gets weaker as prompts become less direct.** Pearson $r$ correlation between response magnitudes (averaged over datasets) measured by direct prompts versus each metalinguistic prompt.

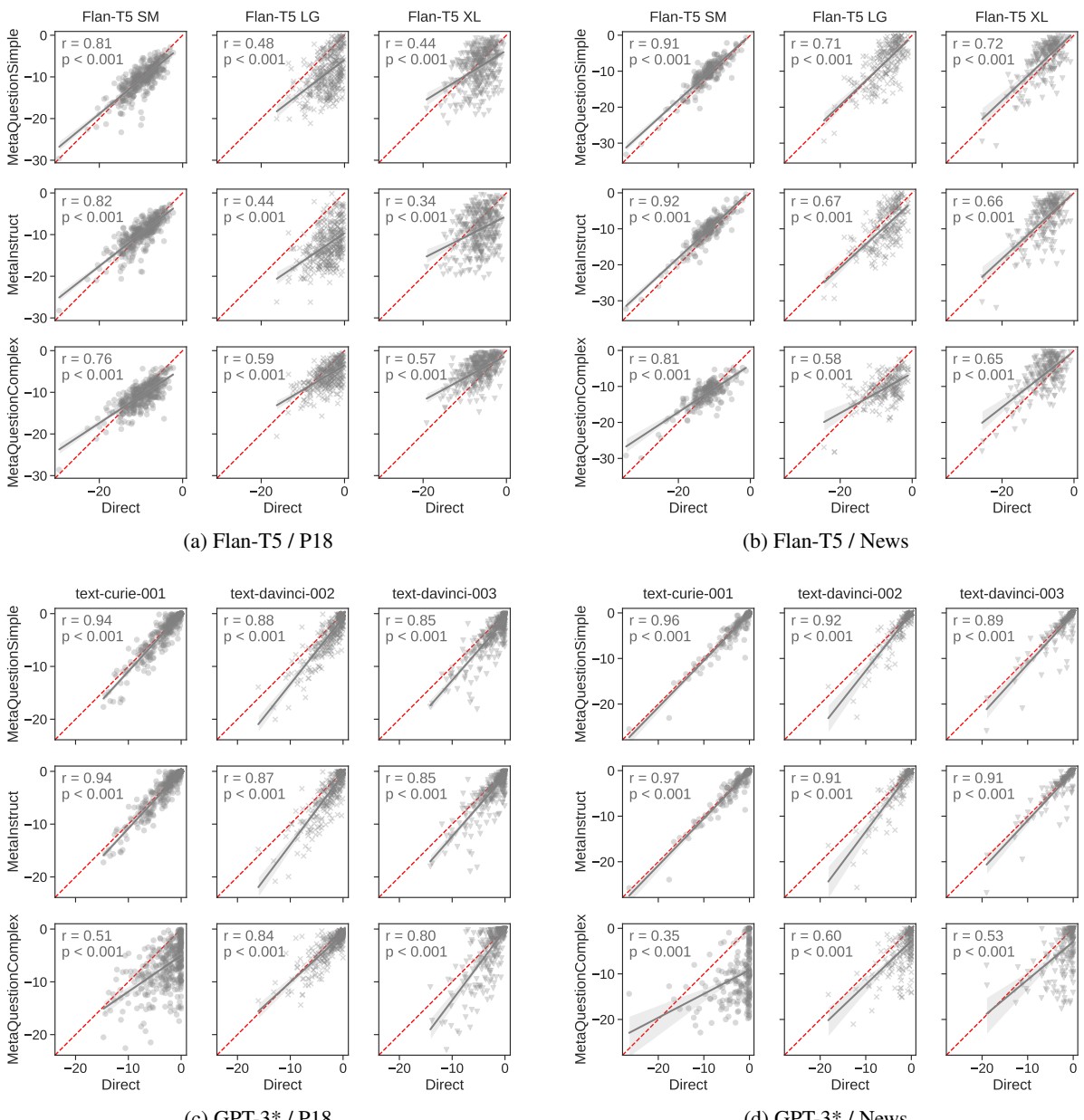

(a) Flan-T5 / P18

(b) Flan-T5 / News

(c) GPT-3* / P18

(d) GPT-3* / News

Figure 7: **Direct vs. metalinguistic responses for Experiment 1.** Relationship between log probability assigned by model to ground-truth sentence continuation under the direct method and each metalinguistic prompting method. Dashed line indicates $x = y$.

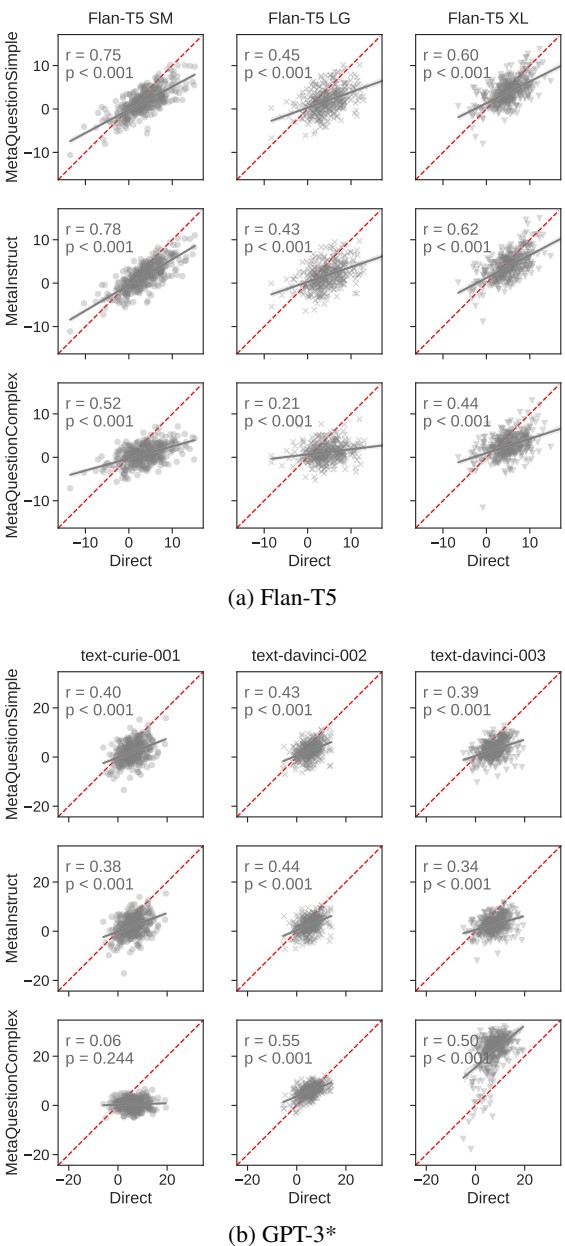

Figure 8: **Direct vs. metalinguistic responses for Experiment 2.** Relationship between log probability differentials (plausible − implausible) measured by direct method and each metalinguistic prompting method. Dashed line indicates $x = y$.

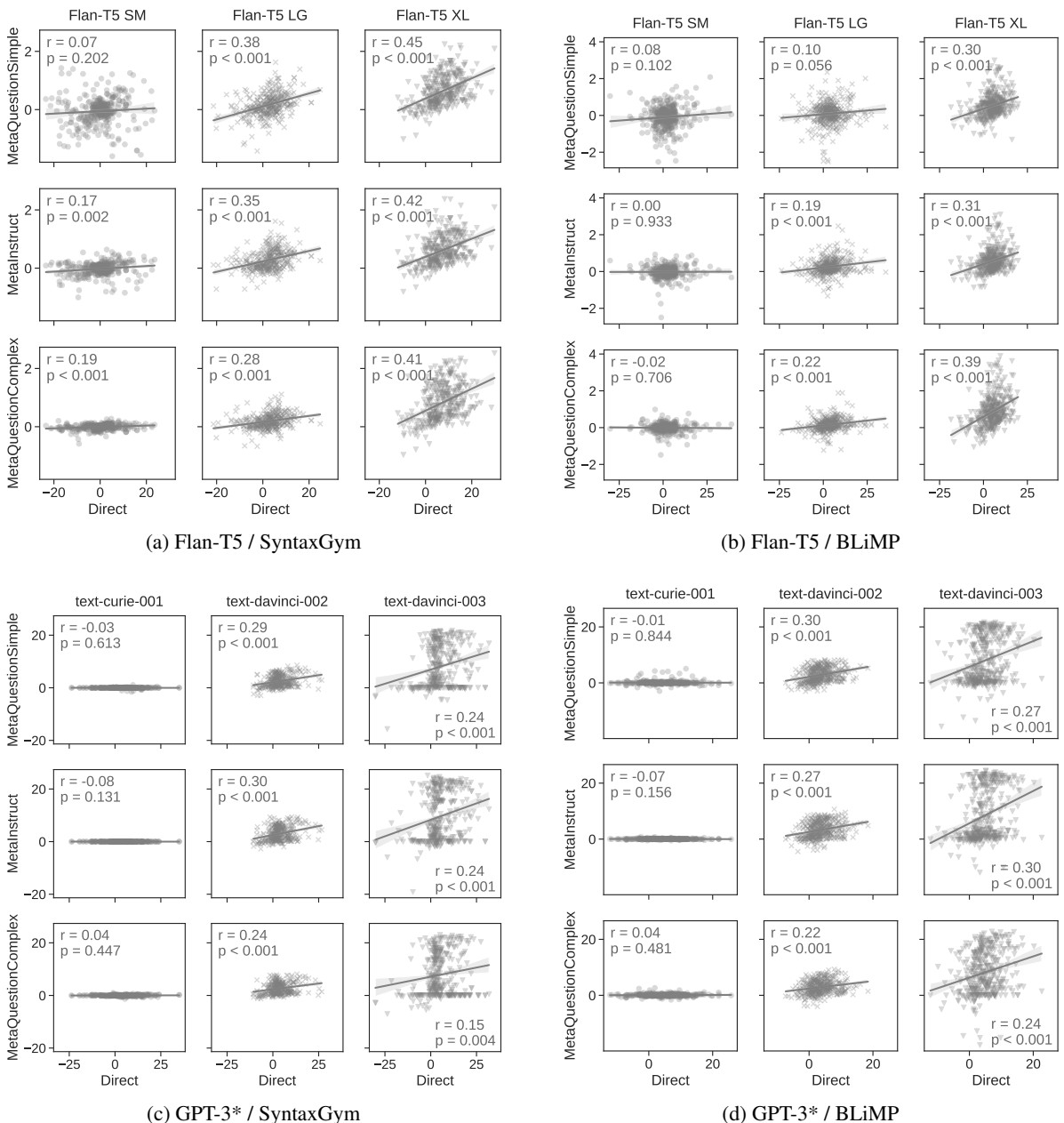

Figure 9: **Direct vs. metalinguistic responses for Experiment 3a.** Relationship between log probability differentials measured by direct method and each metalinguistic prompting method. For the direct method, we compute the difference in log probabilities assigned to the grammatical and ungrammatical sentences. For each metalinguistic prompting method, we compute the difference in log probabilities assigned to the "Yes" token conditioned on the grammatical and ungrammatical sentence prompts (see Section 4.3 for details).

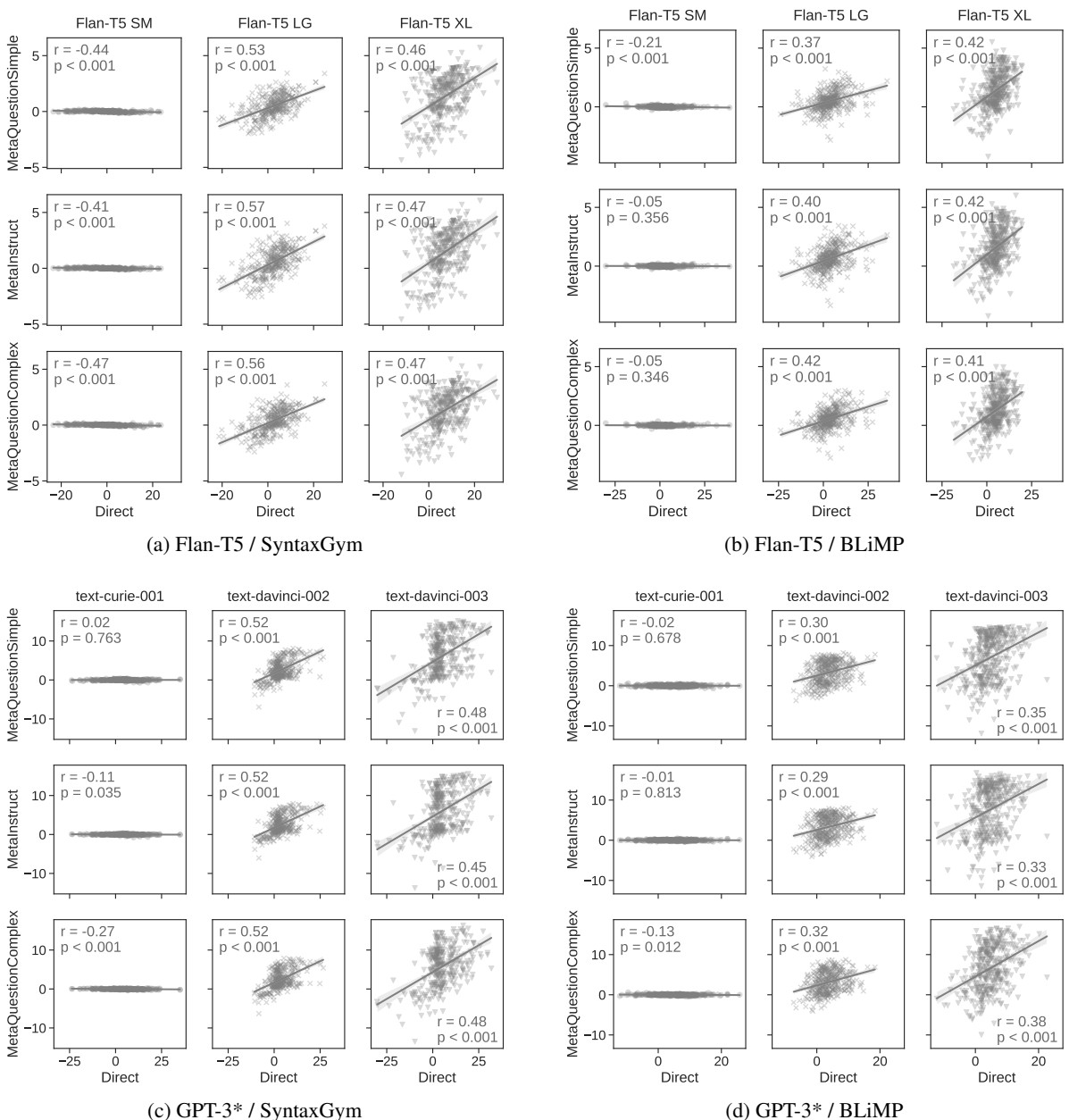

Figure 10: **Direct vs. metalinguistic responses for Experiment 3b.** Relationship between log probability differentials measured by direct method and each metalinguistic prompting method. For the direct method, we compute the difference in log probabilities assigned to the grammatical and ungrammatical sentences. For each metalinguistic prompting method, we compute the difference in log probabilities assigned to the "1" or "2" answer options corresponding to the grammatical and ungrammatical sentences (see Section 4.4 for details).

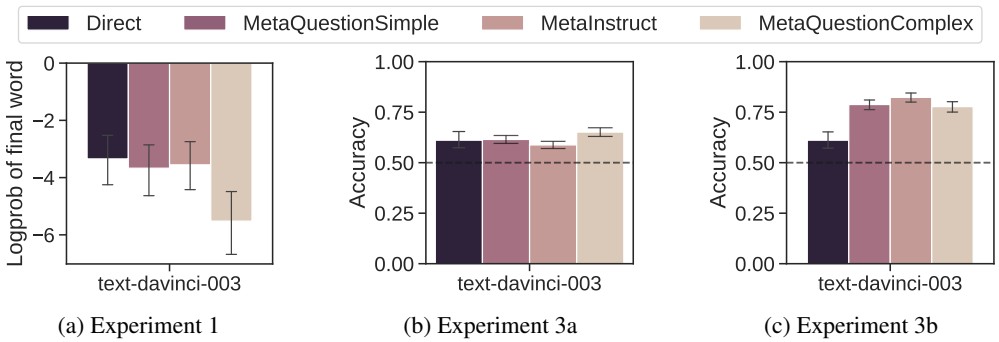

(a) Experiment 1      (b) Experiment 3a      (c) Experiment 3b

Figure 11: Task performance from preliminary experiments in Mandarin Chinese. See Appendix D for details.