# OpenReview forum: "Prompting is not a substitute for probability measurements in large language models"
_EMNLP/2023/Conference — EMNLP 2023 Main_

### Official Review · Reviewer_3kd5 · 2023-08-04

**Soundness:** 4

**Excitement:**

4: Strong: This paper deepens the understanding of some phenomenon or lowers the barriers to an existing research direction.

**Paper Topic And Main Contributions:**

This paper compares prompting and direct evaluation of linguistic knowledge in language models. It shows that evaluation by prompting tests for more than just the ability to encode concepts, and that models generally perform worse on metalinguistic prompts.

My impression is that this is a fairly predictable result but it is valuable to point out and quantify.

**Questions For The Authors:**

What linguistic knowledge is the word prediction task testing, exactly? This should be explained better in the paper imo.

**Reasons To Accept:**

This is a valid and quite thorough evaluation pointing out a relevant limitation of prompting as a way to measure model ability, specifically in linguistic knowledge.

The paper is well-written and the methods are well-described.

**Reasons To Reject:**

- The common wisdom in prompting is to use more on the order of 10 different prompts. I believe it would be helpful, and make for more reliable results, to include more variations of the respective prompt types.
- This is a minor point, but I think the discussion is missing that being able to access this knowledge via prompting may in fact be a desirable property, and should probably be improved.
- The paper repeatedly refers to a debate about “LLMs as models of human language processing” but the connection of such a debate to the work presented is left unclear.

**Reproducibility:**

5: Could easily reproduce the results.

**Reviewer Confidence:**

4: Quite sure. I tried to check the important points carefully. It's unlikely, though conceivable, that I missed something that should affect my ratings.

---

> ### Author Rebuttal · Authors · 2023-08-28
>
> Thank you for your thoughtful comments and feedback. Below, we respond to specific concerns and questions.
>
> We would be happy to include more variations of prompts. To explore this suggestion in the timeframe of the author response period, we tested two additional metalinguistic prompt variations. The first set of new prompts (“MetaInstructKeywords”) is structured such that the relevant sentence (or sentence prefix) and answer are placed at the end of the prompt, with “Sentence:” and “Answer:” keywords. The second set of new prompts (“MetaQuestionComplexOptionFirst”) is structured such that the answer options (only applicable to Experiments 2-3) are presented at the beginning of the prompt, and otherwise mirrors MetaQuestionComplex. (Example prompts are illustrated at the very end of our response.) We re-ran GPT-3.5 (text-davinci-003) on all experiments using these new prompts, and our original qualitative findings still hold. We would also be happy to include more in the final paper, and welcome suggestions for additional prompts to try!
>
> We are also happy to include more discussion on metalinguistic prompting as a potentially desirable property for LMs.
>
> We will also clarify the connection between our work and the debate on “LLMs as models of human language processing”. We mention this debate in order to highlight what’s at stake – i.e., the substantial theoretical claims that researchers are making based on metalinguistic judgments. This helps to motivate our investigation of metalinguistic prompting methods. However, we don’t intend to take a stance on the theoretical debate in our paper.
>
> The word prediction task is not testing a particular component of linguistic knowledge (in the sense of syntax or semantic plausibility) – instead, we treat it as a baseline task. Because we know the model represents next-word-probabilities, we know that the direct probability measurements are “ground-truth”, allowing us to do a tightly controlled comparison to metalinguistic prompting methods. We would be happy to clarify this in the revision.
>
> ---
>
> **New prompt #1: MetaInstructKeywords**
>
> *Experiment 1*
>
> I will show you the beginning of a sentence. Tell me what the most probable next word is.
>
> Sentence: [SENTENCE_PREFIX] …
> Answer:
>
> *Experiment 2*
>
> I will show you the beginning of a sentence. Tell me which is more likely to be the next word of the sentence, from the following options: “arrow”, or “interview”.
>
> Sentence: [SENTENCE_PREFIX] …
> Answer:
>
> *Experiment 3a*
>
> I will show you a sentence. Tell me whether it is a fluent English sentence. Respond with either “Yes” or “No” as your answer.
>
> Sentence: [SENTENCE]
> Answer:
>
> *Experiment 3b*
>
> I will show you two sentences. Tell me which of the two sentences is more fluent in English. Respond with either “1” or “2” as your answer.
>
> Sentence 1: [SENTENCE_1]
> Sentence 2: [SENTENCE_2]
> Answer:
>
> **New prompt #2: MetaQuestionComplexOptionFirst (only for Expts 2-3)**
>
> *Experiment 2*
>
> I will show you the beginning of a sentence, where the next word should either be "arrow" or "interview". [SENTENCE_PREFIX]... Which of these two words is more likely to come next in the sentence?
>
> Answer:
>
> *Experiment 3a*
>
> I will show you a sentence and ask a question with “Yes” or “No” as the answer options. Is the following sentence a good sentence of English? [SENTENCE]
>
> Answer:
>
> *Experiment 3b*
>
> I will show you two sentences and ask a question with “1” or “2” as the answer options. Which of the following two sentences is a better English sentence? 1) [SENTENCE_1] 2) [SENTENCE_2]
>
> Answer:

---

### Official Review · Reviewer_FmJA · 2023-08-04

**Soundness:** 4

**Excitement:**

4: Strong: This paper deepens the understanding of some phenomenon or lowers the barriers to an existing research direction.

**Paper Topic And Main Contributions:**

This paper compares two methods for assessing language models' linguistic knowledge: direct probability measurements and prompting. The tasks on which these methods are compared involve comparing word and sentence probabilities to see how they reflect factors such as semantic plausability and syntactic agreement. Results show that direct probability measurement almost always leads to better performance than prompting. This could potentially motivate future work to prefer linguistic assessments involving direct measurement rather than prompting.

**Questions For The Authors:**

A) Do you have any speculation as to why MetaQuestionComplex performs so well in the text-davinci-002 and text-davinci-003 runs in Experiment 2?

B) Are there cases where you think prompting would still be preferable to direct measurement (assuming both are feasible)?

**Reasons To Accept:**

The experimental results present convincing evidence that direct probability measurement works better for the selected tasks. The methods are clearly outlined, and the inclusion of multiple tasks differing in the directness of prompts is helpful. The paper is well written and easy to follow.

**Reasons To Reject:**

The practical relevance of the findings is not fully clear. My impression is that evaluation related to uncovering linguistic generalizations (e.g. semantic plausibility judgments) is already done using direct probability measurement when possible. The paper did not seem to present examples of tasks where prompting is popular but direct measurement would be better.

Revision: Having read the author response and looked at their cited sources more (e.g., the Katzir and Dentella et al papers), I am now more convinced that prompting is sometimes being used even when direct measurement might perform better. I am therefore increasing my excitement score from 3 to 4.

**Reproducibility:**

5: Could easily reproduce the results.

**Reviewer Confidence:**

3: Pretty sure, but there's a chance I missed something. Although I have a good feel for this area in general, I did not carefully check the paper's details, e.g., the math, experimental design, or novelty.

**Typos Grammar Style And Presentation Improvements:**

In Figure 10(a), the reported correlation values don't seem to match the scatter plots in the left column.

---

> ### Author Rebuttal · Authors · 2023-08-28
>
> Thank you for your thoughtful comments and feedback. Below, we respond to specific concerns and questions.
>
> With regards to practical relevance, there is certainly a substantial body of work using direct probability measurements to evaluate LMs. However, there is also a growing trend to report the outcomes of metalinguistic prompts (e.g., Katzir 2023, Dentella et al. 2023). There will likely be increasing pressures to rely on metalinguistic prompts as our field continues to shift toward models with closed APIs. Therefore, we feel that a detailed investigation comparing the two methods is very timely and relevant, highlighting the value that would be lost if metalinguistic prompting becomes the norm in LM evaluation. We would be happy to clarify this in the revision.
>
> **Question A:** We speculate that this may be because in MetaQuestionComplex for Experiment 2, the answer options (e.g., “arrow, or interview”; see Table 3) are linearly closest to the end of the prompt, where the model begins its generation. To test this idea, we ran GPT-3.5 (text-davinci-003) on a new prompt variant, MetaQuestionComplexOptionFirst, where the answer options are stated near the beginning of the prompt (see response to Reviewer 3kd5 below for more details and examples). We found that the performance on MetaQuestionComplexOptionFirst (0.8) was more similar to the other metalinguistic prompt types, and significantly lower than performance on MetaQuestionComplex (0.96) for Experiment 2. This seems consistent with our speculation. However, the exact causes of variation across prompts are still unclear, and should be investigated in more detail in future work.
>
> **Question B:** Prompting might be useful for evaluating open-ended responses, such as explanations beyond multiple-choice tasks, or for interacting with models in flexible ways, such as chain-of-thought reasoning. Metalinguistic prompting could also allow for more targeted questions that would be challenging to translate into direct probability measurements (e.g., “Which of the following two sentences is more semantically plausible, but less syntactically well-formed?”). We also note that prompting lowers the technical barrier for domain experts outside of NLP to conduct LM evaluations, and could very well be preferred for practical reasons.
>
> **Typos:** Thank you for pointing this out. We chose to keep the x- and y-axis limits consistent through all facets in each subplot (a-d), to facilitate comparison across the facets. It turns out that in the left column of subplot Figure 10(a), most of the y-coordinates points lie within the region [-0.1, 0.1], and thus the points appear to be in a flat line when the y-axis limits are set to [-5, 5]. Apologies for the confusion, and we will do our best to clarify this in the revision.

---

### Official Review · Reviewer_vZ47 · 2023-08-05

**Soundness:** 4

**Excitement:**

4: Strong: This paper deepens the understanding of some phenomenon or lowers the barriers to an existing research direction.

**Paper Topic And Main Contributions:**

Problem or Question Addressed:

The primary concern of the paper is determining the efficacy of metalinguistic prompts in assessing the linguistic capabilities of LLMs compared to direct measurements.It also addresses the issue of whether these two evaluation methods produce consistent results, especially given the move towards closed APIs that may restrict access to direct probability measurements.

Main Contributions:

- Value of minimal-pair comparisons: The paper highlights the significance of using minimal-pair comparisons, a method where two very similar sentences (differing perhaps in only one grammatical aspect) are presented, to reveal models' generalization capabilities.

- Distinction between competence and performance: The study reveals that a model might have the relevant linguistic representation (competence) but may not always be able to access and describe this representation through metalinguistic evaluation (performance). This challenges prior criticisms of LLMs and underscores the difference between what a model knows and how it can express or describe what it knows.

- Implications for LLM evaluations: The study underscores the importance of being cautious about the evaluation methods chosen. It emphasizes that researchers should clearly state their assumptions when selecting a particular evaluation method. Additionally, the study highlights potential challenges and losses in moving towards closed APIs that limit access to probability distributions.

In essence, the paper serves as a cautionary note on how the linguistic capabilities of LLMs are evaluated and the potential pitfalls of relying solely on metalinguistic prompts.

**Questions For The Authors:**

Question A: How generalizable are your findings to language models trained on non-English languages, especially those with significantly different linguistic structures?
Question B: Have your research considered or explored other possible evaluation methods? If not, what are your thoughts on broadening your investigation to include these methods?

**Reasons To Accept:**

The strengths of this paper and the benefits it could offer to the NLP community include:

- Detailed examination of evaluation methods: The paper deeply analyzes how large language models are evaluated, focusing on the differences between metalinguistic prompts and direct measurements. This examination contributes to understanding how well these methods can capture the linguistic competence of the models.

- Practical guidance for researchers: The findings of this study provide invaluable guidance for researchers who work with language models. By highlighting the limitations of metalinguistic prompts, the paper offers insights into how these methods may misrepresent a model's understanding. It encourages researchers to cautiously approach such results, thereby promoting more accurate interpretations of experimental outcomes.

- Theoretical implications: The paper contributes to the broader academic discourse surrounding language models, particularly in relation to the distinction between competence and performance. This theoretical input can stimulate further discussions and investigations in the field.

**Reasons To Reject:**

Some weaknesses of this paper are:

- Limited generalizability: The study primarily focuses on evaluating English language models. This might limit the generalizability of its findings to models trained on other languages, especially those with significantly different linguistic structures.

- Insufficient exploration of alternative approaches: The paper focuses extensively on metalinguistic prompting and direct measurements. Including a wider range of evaluation methods or diversifying the datasets and tasks could provide a more comprehensive understanding.

- Not addressing root causes: While the paper points out the misalignments in LLM responses based on the type of prompt, it doesn't deeply explore the underlying reasons for such discrepancies in LLM behavior.

**Reproducibility:**

3: Could reproduce the results with some difficulty. The settings of parameters are underspecified or subjectively determined; the training/evaluation data are not widely available.

**Reviewer Confidence:**

4: Quite sure. I tried to check the important points carefully. It's unlikely, though conceivable, that I missed something that should affect my ratings.

---

> ### Author Rebuttal · Authors · 2023-08-28
>
> Thank you for your thoughtful comments and feedback. Below, we respond to specific concerns and questions.
>
> **Generalizability beyond English (Question A):** Thank you for raising this point. To explore your suggestion, we performed a preliminary investigation of GPT-3.5 (text-davinci-003) on materials in Mandarin Chinese, which differs from English in many grammatical features. We consulted a native speaker to translate the metalinguistic prompts into Chinese. We then tested GPT-3 on two datasets: (1) a set of recent news articles for word prediction, mirroring Experiment 1; and (2) a set of controlled minimal pairs that cover semantic and syntactic phenomena (Wang, Hu, Levy, & Qian EMNLP 2021 https://aclanthology.org/2021.emnlp-main.454/), mirroring a combination of the phenomena tested in Experiments 2 and 3. We tested the tasks of Experiments 3a and 3b on this set of minimal pairs.
>
> Like our English experiments, in all our Chinese experiments we find that metalinguistic prompting and direct probability measurements do not yield the same results. Like our Experiment 1 in English, we also find that in the Chinese word prediction task, GPT-3.5 assigns highest probability to the ground-truth continuation under the “Direct” method. We also find that model performance improves when using minimal pairs, mirroring our original findings comparing Experiments 3a and 3b. These findings demonstrate how our English results may generalize to languages with different syntactic structures and grammatical properties. However, we also found a difference: in Experiment 3b in Chinese, the “Direct” method underperformed the metalinguistic methods (accuracy scores: Direct ~0.6; others ~0.8).
>
> One potential explanation for this is that according to the OpenAI documentation, the models are “optimized for use in English”, although in practice they may work well for other languages (https://help.openai.com/en/articles/6742369-how-do-i-use-the-openai-api-in-different-languages). Therefore, the models may not be well-suited to scoring probabilities of Chinese sentences with no context (as in the Direct condition); instead, they may benefit from seeing additional Chinese text in the prompt before the sentence (as in the metalinguistic conditions). More broadly, while it would be interesting and useful to run our experiments in other languages, the outputs of the GPT-3/3.5 models in non-English languages may be difficult to interpret.
>
> **Exploring other evaluation methods (Question B):** Thank you for raising this point as well. We choose to focus on prompting and direct probability measurements because we have access to ground-truth probabilities (i.e., by reading out the model logits), which allows us to cleanly test the faithfulness of metalinguistic judgments. In future work, it would be very interesting to compare these approaches to other NLP evaluation methods (e.g., classification tasks like natural language inference). By considering a larger set of methods, the focus would shift away from the validity of metalinguistic prompting, and more toward consistency across methods. For this reason, we feel this warrants a separate follow-up study independent of the current paper.
>
> **Not addressing root causes:** This is a good point, and something that would be very interesting to explore in future work. Future directions could analyze whether models consistently over- or under-report internal probabilities, whether the discrepancies can be predicted from certain prompt structures, or whether certain interventions can increase models’ faithfulness in reporting internal representations.

---

### Meta-Review · Area_Chair_pPtV · 2023-09-15

**Recommendation:** 5

**Metareview:**

All three reviewers provided positive assessments, consistently giving a score of 4 for both soundness and excitement. They consistently agreed that this paper is thorough, well-written, and of clear relevance to the field.

The strengths and weaknesses described by the reviews may be summarized as follows.

Strengths:
- valid and thorough experimental design (R1, R2, R3)
- well-written (R2, R3)
- clear and practical implications for the field (R1)

Weaknesses:
- only tested on one language (English) (R1) and limited exploration of the space of possible prompting strategies (R3)

---

### Decision · Program_Chairs · 2023-10-07

**Decision:**

Accept-Main

**Comment:**

All three reviewers provided positive assessments, consistently giving a score of 4 for both soundness and excitement. They consistently agreed that this paper is thorough, well-written, and of clear relevance to the field.

The strengths and weaknesses described by the reviews may be summarized as follows.

Strengths:
- valid and thorough experimental design (R1, R2, R3)
- well-written (R2, R3)
- clear and practical implications for the field (R1)

Weaknesses:
- only tested on one language (English) (R1) and limited exploration of the space of possible prompting strategies (R3)